# Using Optical Quality Analysis System for predicting surgical parameters in age-related cataract patients

**Thibaud Garcin** [1,2]*, **Damien Grivet**[1], **Gilles Thuret** [1,2,3], **Philippe Gain**[1,2]

**1** Ophthalmology Department, University Hospital, Saint-Etienne, France, **2** Laboratory Biology, Engineering and Imaging of Corneal Grafts, BiiGC, EA2521, Federative Institute of Research in Sciences and Health Engineering, Faculty of Medicine, Jean Monnet University, Saint-Etienne, France, **3** Institut Universitaire de France, Boulevard Saint-Michel, Paris, France

* t.garcin@univ-st-etienne.fr, garcinthibaud@gmail.com

**Data Availability Statement:** All relevant data are within the paper.

**Funding:** The authors received no specific funding for this work.

## Abstract

The Optical Quality Analysis System (OQAS, Visiometrics) provides objective measurements of image formed onto retina, by combining quantification of ocular media transparency and of optical aberrations. In order to evaluate its contribution in the assessment of age-related cataract, we conducted a monocentric clinical study to determine the relationships between clinical grading of lens opacity, OQAS parameters, and parameters required for cataract surgery by phacoemulsification with ultrasound (called "phacodynamics"). Clinical parameters were: best-corrected visual acuity (BCVA, expressed as Log of minimal angle resolution (logMAR)) and the lens opacity classification system III (LOCS III) as a gold standard determined by two independent observers who graded total cataract and nuclear, cortical and posterior sub capsular components. The OQAS provided an objective scatter index (OSI), a modulation transfer function (MTF, expressed in cycle per degree (cpd)) and a Strehl ratio (SR) used as an aberration marker. Patients were operated on by the same surgeon using a phacoemulsification machine that provided the cumulative dissipated energy (CDE) and total ultrasound time (US time) necessary to extract the lens. Patients with poor compliance, corneal or retinal diseases impairing OSI, or who required surgical settings variation, were excluded. Twenty-one eyes of 21 patients aged 76±8 years were analyzed. They were 11 pure nuclear, 3 pure cortical, and 7 mixed cataracts. Mean LOCS III and OSI were respectively: 4.86 ±2.03 and 6.12 ±3.07 (mean±SD). Medians (10˚-90˚ percentiles) were: for BCVA 0.30 (0.10–0.70) logMAR, for MTF cutoff 9.31 (1.54–30.57) cpd, for SR 0.071 (0.042–0.146), for CDE 8.04 (5.74–23.29) and for US time 58 (39–116) seconds. LOCS III was significantly correlated (spearman r, $r_s$) with BCVA ($r_s = 0.561$, p = 0.008), CDE ($r_s = 0.457$, p = 0.038) and US time ($r_s = 0.647$, p = 0.002). The three OQAS parameters significantly correlated (all $r_s \geq 0.526$, p<0.05) with BCVA, and LOCS III grading, but the strongest correlations were found with OSI for cortical components and with MTF for nuclear components: only OSI may be used objectively to assess the effect of cortical components on optical quality, and MTF cutoff—integrating scattering and aberrations—seems the best objective parameter for clinical assessment of nuclear cataracts. The three OQAS parameters were also significantly correlated ($r_s$) with CDE, and with US time only for

**Competing interests:** The authors have declared that no competing interests exist.

pure nuclear cataracts: OSI had the strongest correlations with phacodynamics ($r_s$ = 0.693, p = 0.022 with CDE and $r_s$ = 0.703, p = 0.019 US time). OSI increased with cortical components not requiring higher CDE. When measured in optimal conditions (good compliance, no retinal or ocular surface or tear film diseases), the three OQAS parameters are complementary for objective grading of cataract. In the future, they may help to optimize surgical parameters, especially energy distribution, in femtosecond laser assisted cataract surgery.

## Introduction

Cataract surgery is a major challenge in the world since cataract is the leading cause of blindness and the second-ranking cause of moderate to severe vision impairment [1]. In countries with high standard of living it is the most common surgical procedure, with a constantly increasing number of operated eyes [2, 3]. In France 830,000 cataract surgeries were performed in 2017 [4]. Cataracts are being operated on earlier and earlier due to demographic changes, changing lifestyles, patients' expectations and refinement of surgical techniques.

Health authorities [5–8] and learned societies [9] have long sought to define and regulate decision-making in cataract surgery. Moreover, this surgery is personalized, so it is crucial to define objective criteria that more effectively meet patient needs. In addition, a reliable quantification of lens opacification (accepted as a surrogate criterion for its hardness) could make it possible to optimize the surgical parameters necessary to fragment the lens, whether with ultrasound or with a femtosecond laser (these parameters are hereafter called phacodynamics).

The gold standard gradation system for cataracts remains clinical, based on the Lens Opacities Classification System (LOCS) III [10]. This score, established in 1993 and never revisited, is determined by comparison with a series of reference photographs and separately analyzes the three possible components of a cataract: nucleus (color (NC) and opacification (NO)), cortex (C) and posterior sub-capsular layers (P). Grading should ideally be performed on standardized images, but in practice is most often done by live slit-lamp observation. Despite potential observer and reproducibility biases [11, 12], this is still the scale most widely used to grade cataracts for research or clinical purposes, although there are other systems, such as the Winconsin cataract grading system [13].

New criteria must be explored and integrated in the surgical plan to better meet the expectations of patients with cataract. Various objective techniques have been developed to qualitatively and quantitatively assess lens opacification, which is useful for surgery decision-making. Scheimpflug imaging uses a densitometric analysis that is correlated with LOCS III [14–19] and phacodynamics [16, 19–23] for nuclear cataracts. Lens nucleus density measured by anterior segment optical coherence tomography is also correlated with LOCS III [23, 24]. Wavefront analyzers, measure higher-order aberrations (HOAs), which increase in age-related cataract patients [25–30]. A ray-tracing aberrometry system (iTrace Visual Function Analyzer, Tracey Technologies, Houston, TX) measures a dysfunctional lens index (DLI) in nuclear cataracts [31, 32]. However, none of these techniques measures cataract-induced transparency loss. Shack-Hartmann technology do not consider light scattering [33] and can overestimate the optical quality of an image when scattering affect the eyes, for example in cataract [34].

However, objective measurement of intraocular scattering seems a good way to evaluate the impact of age-related cataract, as this scattering increases with age [35–37]. The Optical Quality Analysis System (OQAS) (HD Analyzer II, Visiometrics SL, Terrassa, Spain) is a double pass system (double pass required to collect light focused on the macula using a 780 nm

infrared laser diode) [38, 39] that provides objective measurements of the image formed onto the retina, by combining the quantification of optical aberrations and of forward plus backward light scattering caused by loss of ocular transparency [34, 40]. The OQAS provides three measures: the objective scatter index (OSI) measures light scattering and optical aberrations [41] by the ratio between integrated light in the peripheral ring and in the area surrounding the central peak of the double-pass point spread function (PSF) image. The OSI scale ranges from 0 (no scattering) to 25 (maximum scattering). The Modulation Transfer Function (MTF) represents the contrast loss in retinal images at various spatial frequencies [37]. The MTF cutoff, in cycle per degree (cpd), is the highest spatial frequency that the eye can detect: the higher the MTF cutoff value, the better the optical quality [42]. The Strehl ratio (SR), for low optic aberrations, is the ratio between the peak intensity from the point spread function (PSF) of the aberrated eye and the peak intensity from the PSF of the unaberrated eye [37, 43]: a higher value indicates better optical quality. The OQAS presents good reproducibility [44] in everyday practice, and repeatability [45] for patients with cataracts or after refractive surgery.

In this research, we analyzed the relationships between the clinical assessment of cataract severity, QAS measurements, and the surgical parameters (energy and time) required to emulsify the lens, in order to analyze the value of the OQAS in cataract surgery planning.

## Material and methods

### Patients and ethic statement

All age-related cataract patients scheduled for surgery under topical anesthesia by a senior surgeon (DG) were included for 30 consecutive days in St Etienne University Hospital. All patients were informed of the nature and intent of the study, and their consent was collected. Study was approved by the local Institutional Review Board "Ethics Committee of the St Etienne University Hospital, Research Commission of Terre d'Ethique" (IORG0007394, IRBN172019/CHUSTE) in accordance with the tenets of the Declaration of Helsinki.

### Preoperative subjective and objective assessments

Complete preoperative examination comprised: preoperative far best-corrected visual acuity (BCVA) measured with routine Monoyer chart (converted to logMAR for analysis), slit lamp evaluation, Goldmann applanation tonometry, fundus examination and optical coherence tomography measurements to exclude macular impairment. Cataracts were graded using LOCS III [10, 11] by two observers (TG/DG), blind to each other, with a slit lamp, after pupil dilation (pupil ≥5mm). Briefly, the three components of cataract were evaluated separately: nuclear opacity (NO) and nuclear color (NC), with scores ranging from 0.1 (clear or colorless) to 6.9 (very opaque or brunescent cataract); posterior sub-capsular (P) and cortical (C), with scores ranging from 0.1 (clear or colorless) to 5.9 (very opaque). The overall score for each cataract (LOCS III total) obtained by adding up the four sub-scores was considered as a continuous variable. Each cataract type was termed pure when no other component was present. Pure nuclear cataracts (N) were quantified by adding NO+NC.

Measurements by the OQAS were done by a single observer (TG) on both eyes of each patient, before any instillation of eyedrops, as recommended by the manufacturer [40, 46], to avoid any measurement bias. The first step was tear film analysis, with OSI measured every 0.5 second during 20 seconds as recommended by the manufacturer, to objectively identify dry eye disease that could prevent from collection of reliable data [47, 48]. Then, data of interest (OSI, MTF cutoff, SR) were acquired by repeating measurements at least six times, as detailed previously by various authors [45, 49, 50] and by calculating their mean without excluding any data.

## Surgical parameters (phacodynamics)

All patients were operated on by the same experienced surgeon (DG) with the same technique, under topical anesthesia by oxybuprocaine chlorhydrate 0,4% and tetracaine 1% (both from Laboratories Théa, Clermont-Ferrand, France) instilled twice in the conjunctival sac, 5 minutes apart, 10 minutes before draping. Surgery was performed through a 2.4mm corneal incision on the 12 o'clock meridian. Dilation was obtained by injecting intracameral Mydrane® 0,2mL (Laboratories Théa). A divide and conquer technique was used with 75% amplitude oZil continuous mode, 10% power phacoemulsification mode, and grade 2 cataract mode on the Infinity® phacoemulsifier (Alcon Laboratories, Fort Worth, Tx, USA). The latter directly provided the cumulative dissipated energy (CDE) and total ultrasound time (US time in seconds). The internal software calculated the CDE in a standardized manner as follows: (torsional amplitude x torsional time x 0.4) + (phaco time x average phaco power).

## Inclusion and exclusion criteria

Of the 28 pre-screened patients, four patients did not meet inclusion criteria as we were unable to obtain optimal OSI measurements on the OQAS [46, 48, 51, 52]: two for poor compliance; one for retinal disease, and one ocular surface diseases (poor-quality tear film).

Twenty-four patients and 26 eyes were included for surgery. Five operated eyes were excluded from analysis to avoid any bias: two eyes for poor cooperation during surgery (thus modifying US time) and two with particularly dense cataract requiring phacodynamics mode adjustment; and one chosen at random for a patient operated on both sides (to avoid statistical bias). In total, we analyzed 21 eyes from 10 women and 11 men with a mean age 76±8 years [range 54–90 years].

## Statistical analysis

Only operated eyes from patients meeting the inclusion and exclusion criteria were studied. Normality of continuous data distribution was analyzed with the Shapiro-Wilk test, with a non-normality threshold set at 5%. Normally distributed data were described by their mean ± standard deviation (SD). Continuous non-normally distributed variables were summarized as median (10˚-90˚ percentiles).

Spearman ($r_s$) correlation coefficients were calculated according to data normality. The sole independent variable was LOCS III grading for cataract severity. All other variables were dependent. Agreement between the two ophthalmologists for LOCS III grading was determined by calculating the mean difference and the correlation coefficient. Partial correlation was used to analyze the relationship between OQAS measurements and phacodynamics parameters controlling for the LOCS III. The relationship between two variables was represented graphically using linear regression. On the linear regression graphs, the red dots show the 95% confidence interval (95% CI) of the slope and intercept. Statistical significance was based on two-tailed statistical analyses, and probability values <0.05 were considered statistically significant, and the beta risk level was 10%. Statistical analyses and graphs were produced using GraphPad Prism 6.0 and SPSS 25.0 IBM Corp, Armonk, NY, USA.

# Results

## Baseline population data

For the 21 operated eyes, median far BCVA was 0.30 (0.10–0.70) logMAR, mean LOCS III total score was 4.86 ±2.03 with excellent agreement between observers (mean difference 0.3; r = 0.94 p<0.001, 19 cases with perfect agreement, three cases with a two-point difference).

**Table 1. Preoperative baseline data (n = 21 eyes).**

| Data | Mean ±SD<br>*Median (10°- 90° percentiles)** | Range |
|---|---|---|
| Far BCVA (BCVA) LogMAR | *0.30 (0.10–0.70)* | 0.10–0.80 |
| Spherical Equivalent | *-0.375 (-5.20 –+1.80)* | -6.5 to +1.875 |
| LOCS III total score (NO+NC+C+P) | 4.86 ±2.03 | 2.00–10.00 |
| LOCS III N score n = 18 | 4.39 ±1.94 | 2–10 |
| LOCS III NO score n = 18 | 2.22 ±1.00 | 1–5 |
| LOCS III NC score n = 18 | 2.17 ±0.99 | 1–5 |
| LOCS III C score n = 10 | 2.00 ±1.16 | 1–4 |
| LOCS III P score n = 1 | 3 | - |
| OSI | 6.12 ±3.07 | 1.40–11.40 |
| MTF cutoff (cycle/degree) | *9.31 (1.54–30.57)* | 1.54–36.45 |
| Strehl ratio | *0.071 (0.042–0.146)* | 0.042–0.148 |

*according to data normality distribution. BCVA = best-corrected visual acuity, LOCS III = lens opacity classification system III, NO = nuclear opalescence, NC = nuclear color, C = cortical opacities, P = posterior subcapsular opacities, OSI = objective scatter index, MTF = Modulation Transfer Function, SD = standard deviation.

There were 18 nuclear, 10 cortical and one subcapsular components. Mean OSI was 6.12 ±3.07, median MTF cutoff was 9.31 cpd (1.54–30.57) and median SR was 0.071 (0.042–0.146). All preoperative population data are summarized in Table 1. Distribution of the components is shown in Table 2: note that each eye may have one or more components. We found 11 pure nuclear cataracts and three pure cortical cataracts.

Regarding phacodynamics, median CDE was 8.04 (5.74–23.29) [2.98–26.47]; median US time was 58 (39–116) seconds [25–122].

## Correlations analysis

**Main objective, subjective preoperative parameters, and phacodynamics.** We found significant correlations between: 1/ BCVA and OSI ($r_s$ = 0.526, p = 0.014), BCVA and LOCS III total ($r_s$ = 0.561, p = 0.008) as well as in subgroups of different LOCS III components (N,

**Table 2. Lens Opacification System III scores distribution, mean values and standard deviation of OSI, median values and percentiles of MTF cutoff and of Strehl ratio (n = 21 eyes).**

| LOCS III score | NO score (n = 18) | NC score (n = 18) | C score (n = 10) | P score (n = 1) |
|---|---|---|---|---|
| 1–1.9 | 4 (22%) | 4 (22%) | 5 (50%) | - |
| 2–2.9 | 8 (44%) | 9 (50%) | 1 (10%) | - |
| 3–3.9 | 5 (28%) | 4 (22%) | 3 (30%) | 1 (100%) |
| 4–4.9 | - | - | 1 (10%) | - |
| 5–5.9 | 1 (6%) | 1 (6%) | - | - |
| 6–6.9 | - | - | | |
| **OSI** Mean ±SD | 5.58 ±2.92 | 5.58 ±2.92 | 7.27 ±3.28 | 3 |
| **MTF cutoff** Median (10˚-90˚ percentiles) | 10.29 (1.54–31.30) | 10.29 (1.54–31.30) | 6.40 (1.54–34.62) | 1.54 |
| **Strehl ratio** Median (10˚-90˚ percentiles) | 0.077 (0.042–0.147) | 0.077 (0.042–0.147) | 0.059 (0.042–0.144) | 0.042 |

LOCS = lens opacity classification system, NO = nuclear opalescence, NC = nuclear color, C = cortical opacities, P = posterior subcapsular opacities, OSI = objective scatter index, MTF = Modulation Transfer Function, SD = standard deviation.

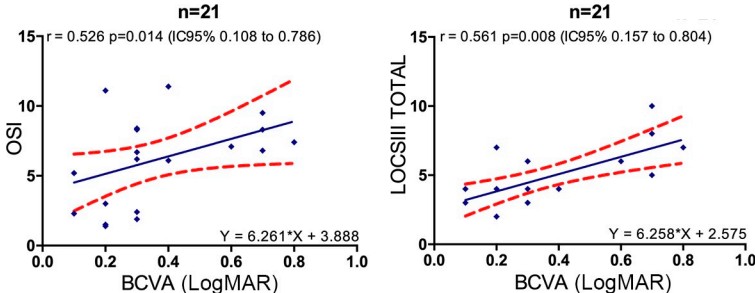

**Fig 1. Linear regressions between BCVA and OSI, and between BCVA and LOCS III total.** Red dots show the 95% CI of the slope and intercept. OSI = objective scatter index, BCVA = far logMAR best-corrected visual acuity, LOCS III = lens opacity classification system III.

NC, NO with significant $r_s = 0.633$, $r_s = 0.621$, $r_s = 0.581$ respectively) except for LOCS III C ($p = 0.696$) (Fig 1); 2/ LOCS III total and OSI ($r_s = 0.586$, $p = 0.005$) as well as in subgroups of different LOCS III components (N, NC, NO, C) (Fig 2); 3/ LOCS III total and CDE ($r_s = 0.457$, $p = 0.038$) as well as in subgroups of different LOCS III components (N, NC, NO) except for LOCS III C ($p = 0.052$); LOCS III total and US time ($r_s = 0.647$, $p = 0.002$) as well as in subgroups of different LOCS III components (N, NC, NO) except for LOCS III C ($p = 0.054$) (Fig 3).

For pure nuclear cataracts only, CDE increased significantly with OSI ($r_s = 0.693$, $p = 0.022$), as well as total US time with OSI ($r_s = 0.703$, $p = 0.019$) (Fig 4). No significant correlation was found between OSI and CDE if N, C, P components were taken together for analysis. Partial correlation between CDE and OSI, US time and OSI controlling for the LOCS III was negligible ($r = 0.013$ and $r = -0.041$ respectively) and not significant ($p = 0.993$ and $p = 0.910$

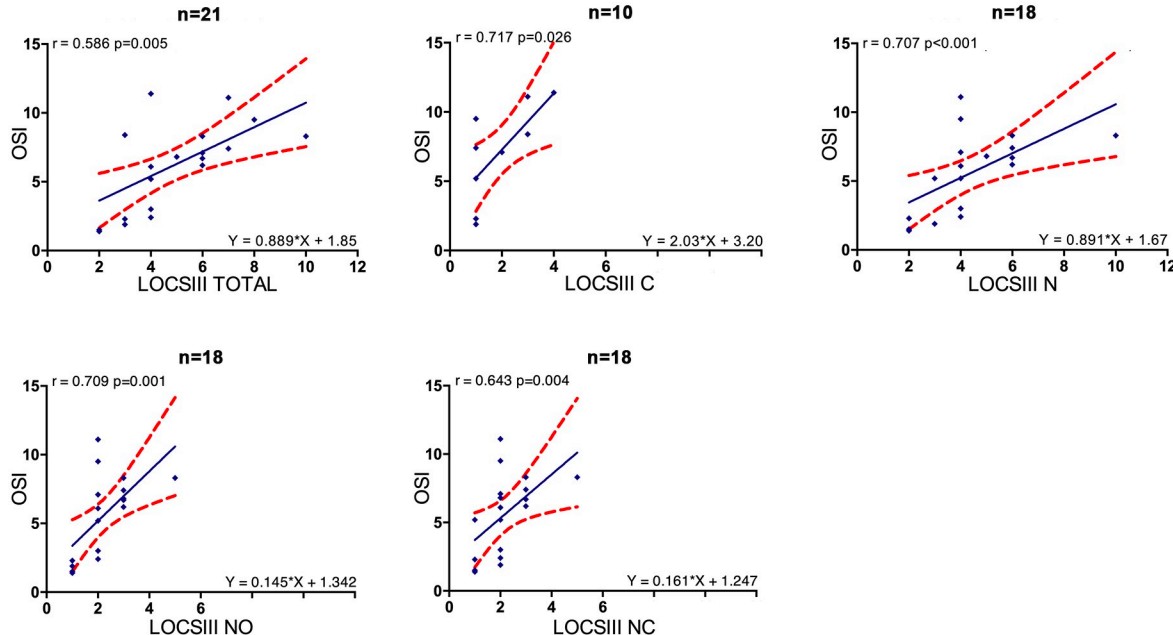

**Fig 2. Linear regressions between OSI and LOCS III.** Red dots show the 95% CI of the slope and intercept. OSI = objective scatter index, LOCS III = lens opacity classification system III, N = nuclear component = (NO+NC), NO = nuclear opalescence, NC = nuclear color, C = cortical opacities.

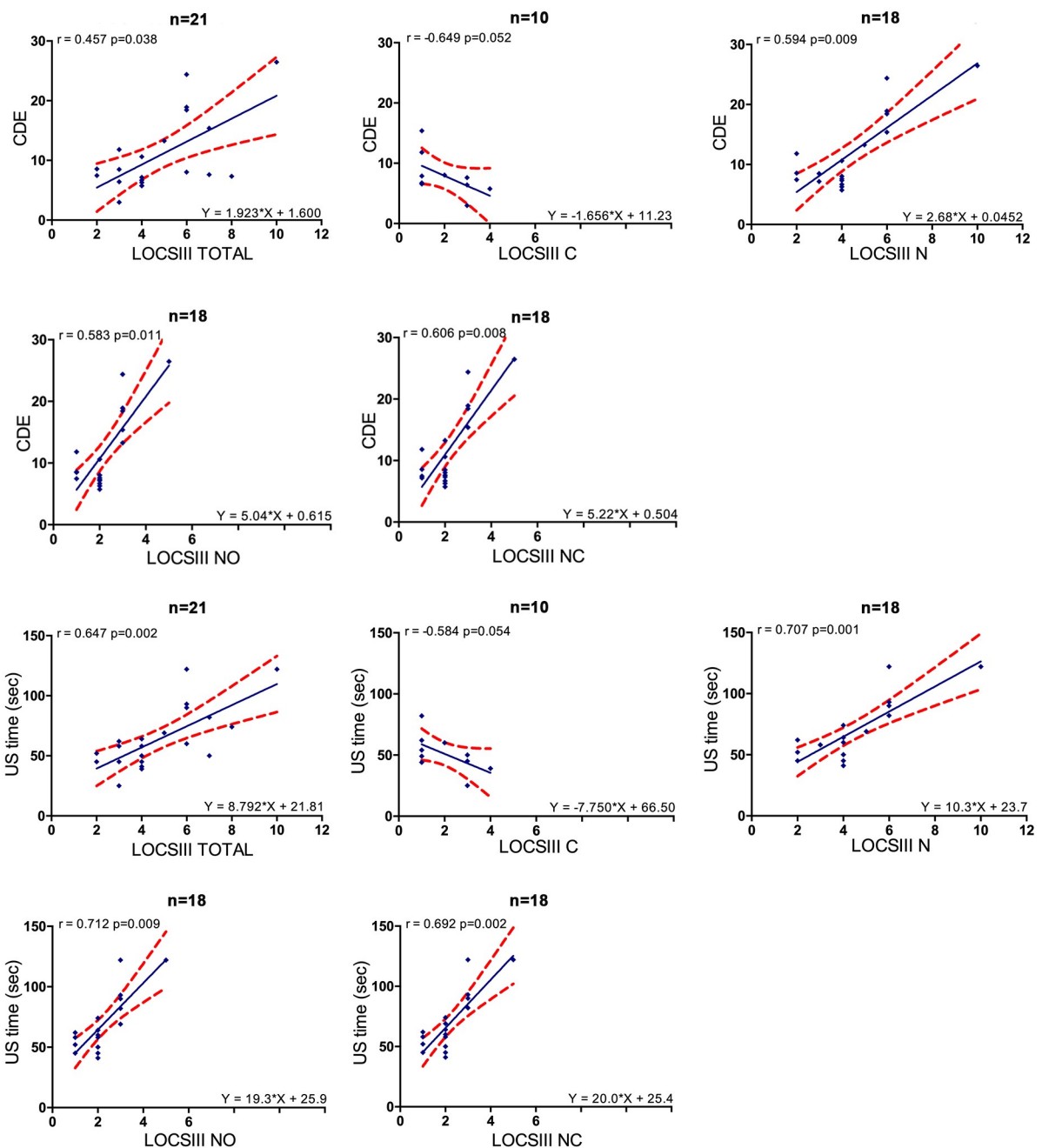

**Fig 3. Linear regressions between LOCS III and surgical parameters.** Red dots show the 95% CI of the slope and intercept. CDE = cumulative dissipated energy, US = ultrasound, LOCS III = lens opacity classification system III, N = nuclear component = (NO +NC), NO = nuclear opalescence, NC = nuclear color, C = cortical opacities.

respectively), thus indicating that LOCS III N score greatly influenced the relation between OSI and surgical parameters.

**Other OQAS parameters, subjective preoperative parameters and phacodynamics.** Strong significant correlations were found between: 1/ BCVA and MTF cutoff ($r_s$ = -0.685, p<0.001); 2/ LOCS III total and MTF cutoff ($r_s$ = -0.711, p<0.001) as well as in subgroups of different LOCS III components (N, NC, NO with significant $r_s$ = -0.771, $r_s$ = -0.675,

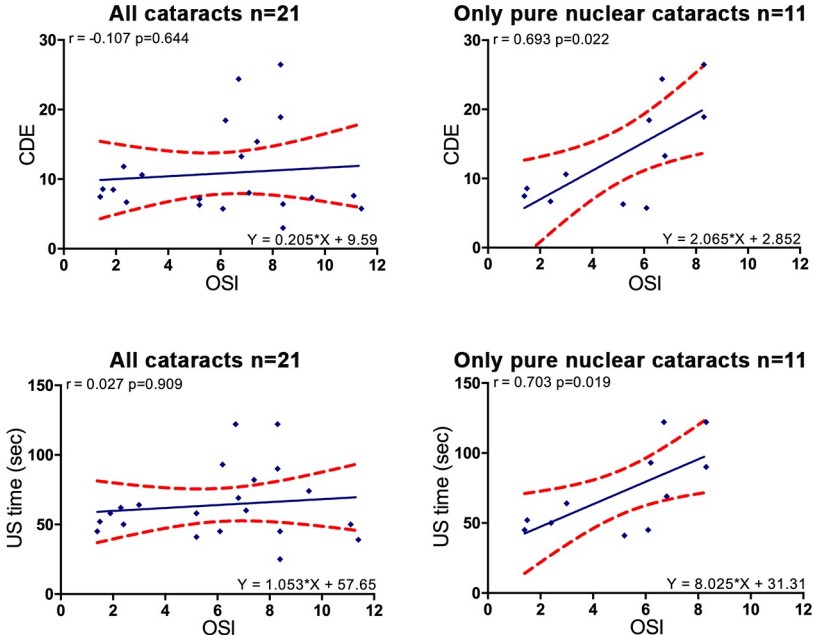

**Fig 4. Linear regressions between OSI and surgical parameters for all cataracts and for pure nuclear cataracts.** Red dots show the 95% CI of the slope and intercept. OSI = objective scatter index, US = ultrasound, CDE = cumulative dissipated energy, LOCS III = lens opacity classification system III, N pure = pure nuclear cataract.

$r_s$ = -0.762 respectively) except for LOCS III C (p = 0.186); 3/ MTF cutoff and CDE ($r_s$ = -0.655, p = 0.034), and also MTF cutoff and US time ($r_s$ = -0.648, p = 0.034) only for pure nuclear cataracts (Fig 5).

We also found strong significant correlations between: 1/ BCVA and Strehl ratio ($r_s$ = -0.622, p = 0.003); 2/ LOCS III total and Strehl ratio ($r_s$ = -0.676, p<0.001) as well as in subgroups of different LOCS III components (N, NC, NO with significant $r_s$ = -0.745, $r_s$ = -0.640, $r_s$ = -0.758 respectively) except for LOCS III C (p = 0.056); 3/ Strehl ratio and CDE ($r_s$ = -0.664, p = 0.031), and also Strehl ratio and US time ($r_s$ = -0.653, p = 0.032) only for pure nuclear cataracts (Fig 6).

## Discussion

In our study, the OQAS was examined to assess its usefulness in clinical practice and try to answer if it may be a reliable tool for clinical objective assessment of different age-related cataract types, but also for prediction of phacodynamics.

Our population sample was representative of general French population who undergo cataract surgery [4]. Even if BCVA may be less considered now, as the latest recommendations not include it in surgery decision-making, our mean BCVA respected the previous cutoff established by the competent Health Authorities, like Hwang et al. [53], or contrary to some studies evaluating the OQAS on patients with lower preoperative logMAR BCVA at earlier stages age-related cataract [21, 49, 50, 54–56].

As a gold standard for cataract grading, we used the LOCS III classification, as in the initial publication [10] and obtained a strong agreement between both observers and reliable classification [10–12]. The level of LOCS III for surgery decision-making has decreased over time since 1993 [50, 56–58] which limits studies' comparability. In addition, in the first OQAS study, Artal et al. [40] interpreted LOCS III classification by redefining three LOCS III

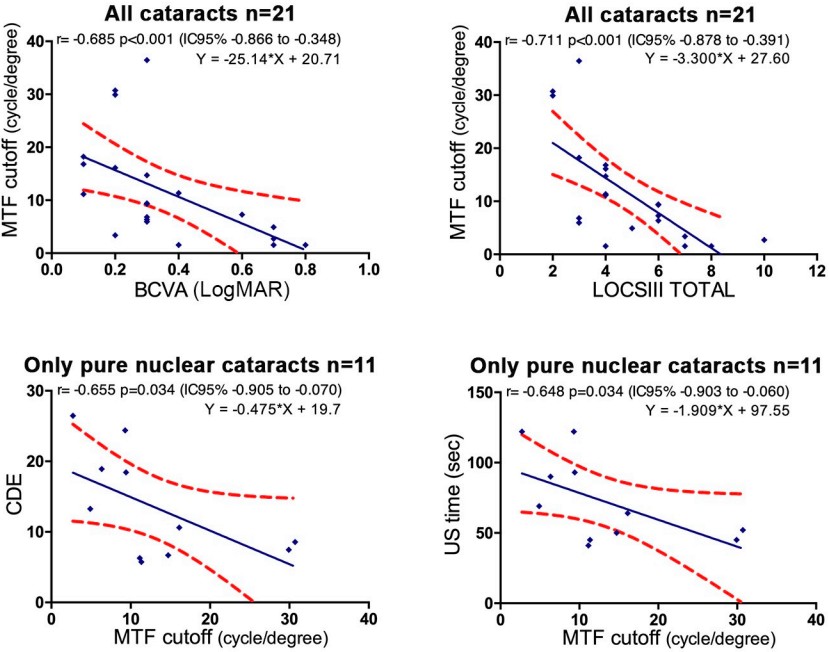

**Fig 5. Linear regressions between MTF cutoff and clinical or surgical parameters.** Red dots show the 95% CI of the slope and intercept. MTF = Modulation Transfer Function, BCVA = far logMAR best-corrected visual acuity, LOCS III = lens opacity classification system III, CDE = cumulative dissipated energy, US = ultrasound.

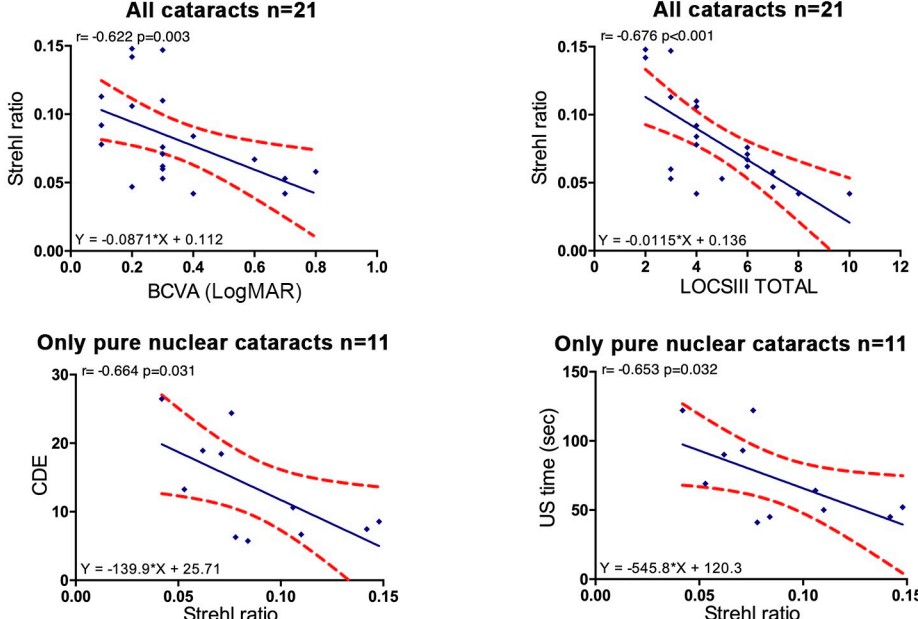

**Fig 6. Linear regressions between Strehl ratio and clinical or surgical parameters.** Red dots show the 95% CI of the slope and intercept. BCVA = far logMAR best-corrected visual acuity, LOCS III = lens opacity classification system III, CDE = cumulative dissipated energy, US = ultrasound.

subgroups for nuclear cataracts, as Vilaseca et al. [59] did for the three cataract components (N/C/P), limiting comparability between papers on clinical grading of cataracts.

As nuclear and cortical components were well represented in our population, correlations between the LOCS III total and BCVA confirmed results from previous studies [14, 21, 50]. LOCS III remains convenient, and cost-effective to assess the impact of nuclear cataract on visual acuity. This is not the case for the cortical components because all opacities can be graded since there are located at the periphery, whereas BCVA decreased only when central cortex (generally the central 4.0mm) is concerned by moderate to advanced cortical cataracts.

As previously described [23, 60, 61], we logically found moderate to strong correlations only between LOCS III grading of nuclear components and phacodynamics. Contrary to the lens nucleus that becomes harder when it loses transparency with age (due to centripetal compaction of lens fibers), the cortex, composed of loose fibers, remains soft when it becomes opaque and does not require increased ultrasound energy nor time (only irrigation-aspiration is usually used).

Previous studies on the OQAS showed no consensus on population selection: most authors chose to include cataracts with different components (N/C/P) [49, 50, 53, 55, 56, 59], but one selected only nuclear cataracts [21]. As cortical components cause disability in daily life without necessarily greatly impairing visual acuity [57, 62], we chose to consider every type of cataract (N/C/P) for both population selection and LOCS III grading, to prevent potential bias: Artal et al. [40] did not consider this point in selecting their population, and mentioned it as a limitation. Other authors analyzed correlations for nuclear cataracts only [50], or considered all components as nuclear cataracts, judging that other types of cataracts were poorly represented in their study [54].

The principle of OQAS measurement requires a healthy eye surface. We have therefore excluded dry eye syndromes that affect the passage of light, to provide reliable objective data as recommended [45, 46, 49, 51].

We found moderate correlation between OSI and BCVA, like other teams [49, 56], while other authors found a stronger correlation [21, 50, 59]. Differences can be explained by centers' different visual acuity thresholds for surgery decision-making, and, as explained above for correlation between LOCS III and BCVA, by the different proportions of nuclear and cortical cataracts [59], cortical components being well represented with different grades in our population.

Similarly, we found moderate to strong correlations between OSI and different components of LOCS III. Again, variations exist with previous studies which found stronger [21] or weaker correlations [50, 54, 56] probably because of differences in spectrum of cataract severity.

Until now, among the three measurements provided by the OQAS, OSI is the most frequently studied and seemed the most effective one for cataract surgery decision-making with high specificity and sensibility [53, 55], with OSI from 3 to 7 deemed a good indication for surgery when all cataract components (N/C/P) are considered [40, 50]. Notably, OSI is influenced by low or high order aberrations [40], which explains why at least spheric and cylindric ametropias must be corrected before OQAS measurement. In the present study, we also analyzed the MTF cutoff and the SR.

The MTF cutoff and the SR values decreased significantly as OSI increased, even more than previously reported [50]. The MTF cutoff was strongly correlated with LOCS III N, with higher association than previously described [50, 63]. Cortical components are known to increase OSI [49, 55, 59]. Interestingly, we found no correlation between MTF cutoff and LOCS III C or SR and LOCS III C (more sensitive to pupil diameter), whereas OSI increased strongly with LOCS III C. We hypothesize that variations in cataract severity between studies could explain this difference, as some of our patients, had visible cortical opacities in the 4.0mm pupillary

area. In total, by combining the three parameters, OQAS thus makes it possible to separate the impact of moderate to severe cortical cataract components on visual quality. In contrast, compared to OSI or SR, MTF cutoff had the strongest correlations with BCVA, LOCS III N, NO, NC. As MTF cutoff simultaneously considers scattering and aberrations [49, 63], MTF cutoff may be the best objective OQAS parameter for clinical assessment of nuclear cataracts.

Reliable prediction of phacodynamics would considerably help to optimize cataract surgery devices (ultrasound or femtosecond laser), avoiding over- and underestimations due to preset parameters, which are both detrimental. We chose CDE as the main criterion because it is the most relevant data currently used to quantify US energy as used to emulsify the lens [64].

Importantly, our study reports a new key point: when considering mixed cataract types, no correlation exists between OQAS measurements (OSI, MTF cutoff, SR) and phacodynamics (CDE or US time). OSI increased with cortical components that not requiring more CDE or US time. After removing cortical and sub capsular components, for pure nuclear cataracts only, OSI ranged from 1.4 to 8.3 and strong correlations were found between CDE and OSI. The correlation was even stronger than between CDE and LOCS III N, suggesting that OQAS could better predict phacodynamics than LOCS III for pure nuclear cataracts. The same findings were found for US time and OSI, and for US time and LOCS III N. Partial correlation analysis clearly demonstrates that LOCS III acts as a confounding factor to explain the strong correlation between OSI and phacodynamics. This is because LOCS III and the OQAS both measures the same physical phenomenon -lens opacification- if there are no further ocular pathologies. This demonstrates OQAS reliability as objective and automatic measurement. In addition, OSI had stronger correlations with phacodynamics than MTF cutoff or SR, suggesting that OSI may be the OQAS parameter for optimizing surgical plans with energy modulation. Correlation between OSI and CDE was also previously studied [21] as secondary endpoint, with several limitations including: selection only of cataracts with a predominant nuclear component without stating whether there were pure nuclear cataracts, contrary to Artal et al. [40]; absence of details on surgical protocol, suggesting possibly variation in preset program and thus in energy levels; and no clear exclusion of ocular surface diseases, which may change OQAS measurements [47, 48, 52].

Our research also has limitations: 1/ although our sample is representative of patients typically operated for age-related cataract in many centers worldwide, it is small, limiting the range of LOCS III scores. No statistic could be generated for subcapsular posterior cataracts (n = 1). Lack of statistical significance for correlation between LOCS III C and CDE as well as US time may be due to the insufficient number of patients in this subgroup; 2/ despite being the clinical gold standard, LOCS III does not differentiate peripheral from central opacities, resulting in discordance between symptoms and the LOCS III score [62]. Therefore, it may be more suitable to precisely assess and consider cortical components with another device or with another clinical scale, such as the Wisconsin score system.

## Conclusions

The OQAS improves preoperative assessment of age-related cataract patients. We confirm that OSI, measured in optimal conditions (good compliance, no retinal or ocular surface or tear film diseases), is correlated with clinical parameters, but MTF cutoff, incorporating scattering and aberrations, seems the better objective parameter to assess nuclear cataracts. Only OSI may be used to assess objectively the effect of cortical components on optical quality. OSI may also predict the phacodynamics (US energy and time) needed, but only for pure nuclear cataracts, as OSI increased with cortical components that do not require more CDE or US

time. The OQAS provides objective and automatic measurements that can be used to personalize cataract surgery parameters.

## Author Contributions

**Conceptualization:** Thibaud Garcin, Damien Grivet, Gilles Thuret, Philippe Gain.

**Data curation:** Thibaud Garcin, Damien Grivet.

**Formal analysis:** Thibaud Garcin, Gilles Thuret.

**Funding acquisition:** Gilles Thuret, Philippe Gain.

**Investigation:** Thibaud Garcin, Damien Grivet, Gilles Thuret, Philippe Gain.

**Methodology:** Thibaud Garcin, Damien Grivet, Gilles Thuret, Philippe Gain.

**Project administration:** Thibaud Garcin, Gilles Thuret, Philippe Gain.

**Resources:** Thibaud Garcin, Damien Grivet, Philippe Gain.

**Software:** Thibaud Garcin, Gilles Thuret.

**Supervision:** Thibaud Garcin, Damien Grivet, Gilles Thuret, Philippe Gain.

**Validation:** Thibaud Garcin, Damien Grivet, Gilles Thuret, Philippe Gain.

**Visualization:** Thibaud Garcin, Damien Grivet, Gilles Thuret, Philippe Gain.

**Writing – original draft:** Thibaud Garcin.

**Writing – review & editing:** Thibaud Garcin, Gilles Thuret, Philippe Gain.

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
