## [Decision Letter · Decision Letter 0]

7 May 2020

PONE-D-20-08844

Using Optical quality analysis system for cataract preoperative assessment & phacodynamics prediction in age-related cataract patients

PLOS ONE

Dear Dr. Garcin,

Thank you for submitting your manuscript to PLOS ONE. After careful consideration, we feel that it has merit but does not fully meet PLOS ONE’s publication criteria as it currently stands. Therefore, we invite you to submit a revised version of the manuscript that addresses the points raised during the review process.

We would appreciate receiving your revised manuscript by Jun 21 2020 11:59PM. To enhance the reproducibility of your results, we recommend that if applicable you deposit your laboratory protocols in protocols.io, where a protocol can be assigned its own identifier (DOI) such that it can be cited independently in the future. For instructions see: http://journals.plos.org/plosone/s/submission-guidelines#loc-laboratory-protocols

We look forward to receiving your revised manuscript.

Kind regards,

Ireneusz Grulkowski, PhD

Academic Editor

PLOS ONE

Journal Requirements:

"NO

The funders had no role in study design, data collection and analysis, decision to

publish, or preparation of the manuscript."

Reviewers' comments:

Reviewer's Responses to Questions

**Comments to the Author**

1. Is the manuscript technically sound, and do the data support the conclusions?

Reviewer #1: Partly

Reviewer #2: Yes

Reviewer #3: Partly

2. Has the statistical analysis been performed appropriately and rigorously? 

Reviewer #1: No

Reviewer #2: Yes

Reviewer #3: Yes

3. Have the authors made all data underlying the findings in their manuscript fully available?

Reviewer #1: Yes

Reviewer #2: Yes

Reviewer #3: Yes

4. Is the manuscript presented in an intelligible fashion and written in standard English?

Reviewer #1: No

Reviewer #2: Yes

Reviewer #3: No

5. Review Comments to the Author

Reviewer #1: The study evaluated the association between OSI, LOCS III and phacodynamics, and there are some questions need to be considered.

1. The title used the Optical quality analysis system, but the author seems to focus on the OSI. The criterion was only related to OSI, so I am not clear why the relationships between MTF cutoff, LOCS III and phacodynamics were studied here. The abstract also did not mention the results of MTF cutoff.

2. The English used in the paper need to be improved. Occasionally, it is difficult to understand some statements. A professional editing service is urgently needed.

3. The Introduction part is relatively long, and the author should simplify the text.

4. The cataract grading score (LOCS III) used in the study were grade variables, which should not use mean±SD, and the correlation analysis should be changed to spearman correlation. In addition, the author mentioned the regression analysis in the Figure lengends, but not occurred in the main manuscript. And the independent variables and dependent variables should be defined in the whole study.

5. The author analyzed the correlation between OSI and LOCS III, LOCS III and phacodynamics, then analyze the association between OSI and phacodynamics. The Partial correlation need to be used and the LOCS III should be considered in the association between OSI and phacodynamics.

6. The “95%CI” in Figure legends should be cleared.

Reviewer #2: Dear Thibaud Garcin,

First of all, I would like to thank for submitting in PLOS ONE your work “Using Optical quality analysis system for cataract preoperative assessment & phacodynamics prediction in age-related cataract patients”. I have to admit that this work might be very useful for ophthalmologists and consequently, it might improve the life quality of our society. However, there are some doubts, that I would like to ask you:

Material and Methods

a) Involved eligible patients.

• In relation to the topic “involved eligible patients”, it is indicated that all patients were under topical anaesthesia. Is it possible to indicate from which brand was used this anaesthesia?

• In this section, it might appropriate to include information about the subjects who participated in this clinical trial (initial number of patients and eyes, age and gender, refractive error, etc.) instead of the results section.

b) Preoperative subjective and objective assessment.

• In relation to the gold-standard LOCSIII test, I would like to know the reason why the nuclear opacity (NO) and nuclear colour (NC) are scaled from 1 to 6 meanwhile the sub-capsular (SCP) and cortical cataract (C) are scaled from 1 to 5. Moreover, it might be very useful to indicate it in the manuscript.

• OQAS measurements were done in both eyes where some subjects showed that one eye was undergone to phacoemulsification. This data allowed to check the reliable and consistent of the OQAS. It might be very interested to include these to show the advantage of the OQAS.

c) Statistical analysis

• Regarding to the statistical analysis section, it is not indicated which type of test is used to compare the proposal parameters. I would recommend to indicate that all parameters were compared by linear correlations in order to avoid misunderstanding with another tests like T – student test or ANOVA.

• In the line 210, it is indicated “mean + SD”. Is it possible to indicate in the manuscript what it means SD.? I think, it is Standard Deviation.

Results

a) Baseline population data

• LOCSIII test was done by two observers and no differences were appreciated between them. I would like to ask you how it was analysed the reproducibility of the LOCSIII between these 2 observers to obtain a p-value = 0.51.

• In table 1, is it possible to indicate which parameters are following a normal and not normal distribution after doing the Sapphire – Wilk test. In addition, in the title box Mean +- SD, it should be included median +- percentiles.

b) Correlations

• In relation to figure 2, I am quite surprised about the dispersion when it is correlated the BCVA with the OSI and with LOCSIII test. I am wondering if this dispersion might be associated to the low influence of the visual acuity in the decision to cataract surgery as it was indicated in the introduction. It would be fantastic to discuss this point in the discussion section.

• Another interesting point that I would like to ask you in relation to the figure 3 is about the slopes of each graphic. It seems that the correlation LOCSIII C vs OSI shows the higher slope (with r=0.715, which it is the highest), but as well the most dispersive graphic (p-value = 0.02 although <0.05). How clinically might it be interpreted? It is a pity that the sample population is too low (n=10) to analyse better this behaviour. Moreover, according to these graphics, it is observed that the scattering is higher in nuclear cataracts, but the strong OSI change is caused in the cortical cataracts? Do you know why?

• Regarding to figure 4, how is it possible that the CDE hasn´t got any influence in the cortical cataract? According to your correlations, it is not significance (p-value = 0.058). I think, that it is caused by the sample population n=10. What do you think?

• According to figure 6, I think it does not have any sense to include it because the MTF and OSI are specific parameters from OQAS which are strongly related between them by the PSF. Thus, the OQAS must give strong correlation to have a high reproducibility. Consequently, this correlation does not give any significant information for a clinical purpose. I would recommend to remove the data and graphics between MTF and OSI and prepare another figure with correlations between MTF with BCVA, MTF with LOCSIII, MTF with CDE and MTF with US.

Discussion

• In the discussion about the BCVA vs OQAS, it is explained that the differences between your research with other ones might be caused by the sample size or the visual acuity decision-making surgery after consulting people. However, there is not any references about the type of visual test was used and how it might impact in the correlated results.

Reviewer #3: COMMENT:

This article discusses the usefulness of the OSI parameter in the clinical management of age-related cataracts. Furthermore, the relationship of preoperative parameters with photodynamic parameters involved in surgery is interesting and original. However, the sample of patients included in the study is very small. In addition, as several works recommend, the statistical analysis must be corrected to include 1 eye per patient. In fact, only one case in this work does not meet this criterion and the results of the study are unlikely to be different if the correction is made. Likewise, the discussion should be improved for a better understanding. For example, when the author removes cases with a cortical component, he consequently removes cases with OSI> 9. This should be added in the discussion. The influence of aberrations on the overestimation of OSI should be discussed as well.

As far as language and style are concerned, the use of punctuation marks does need some revising as on certain occasions it interferes with clarity. See, for example, lines 341 – 345 and 402 - 403. Similarly, the paper shows some segments lacking in clear sentence structure that should be rewritten (lines 21 – 23; 159 -163; 207 – 208; 222 – 224). Other aspects that should be revised include unnecessary use of passive forms - particularly of the verb “correlate” -, the use of contractions, vocabulary (“similar as” in lines 364 – 385 should be “similar to”), missing words (“decision-making surgery”) and verb tense consistency (lines 160 – 161).

SPECIFIC COMMENTS:

Page 8. Lines 159-163: this sentence should be rewritten for a better understanding.

Page 9. Lines 180-183: the definition of SR does not exactly correspond to the one used by the double-pass system, which computes the SR in two dimensions as the ratio between the areas under the MTF curve of the measured eye and that of the aberration-free eye. Ref.: DOI:10.1111/j.1444-0938.2010.00535.x

Page 9. Lines 194-195: missing parenthesis open and close.

Page 11. Line 225: in the statistical analysis it is recommended to include an eye per patient in all cases due to strong correlation between the two eyes of a subject.

Page 14. Lines 298-303: correlations with SR should be included in the results since this parameter is the one most related to loss of image quality and very sensitive to aberrations.

Page 14. Line 305: correlation between the MTF and OSI is evident, especially considering that they have been obtained with the same instrument. Therefore, Figure 6 is unnecessary.

6. PLOS authors have the option to publish the peer review history of their article (what does this mean?). If published, this will include your full peer review and any attached files.

Reviewer #1: No

Reviewer #2: No

Reviewer #3: No

---

## [Author Response · Author response to Decision Letter 0]

8 Jul 2020

PONE-D-20-08844

Using Optical quality analysis system for cataract preoperative assessment & phacodynamics prediction in age-related cataract patients

PLOS ONE

Saint-Etienne, July 4th 2020

Dear Ireneusz Grulkowski,

Please find attached a revised version of our article titled “Using Optical quality analysis system for cataract preoperative assessment & phacodynamics prediction in age-related cataract patients” with a new title "Using Optical Quality Analysis System (OQAS) for predicting surgical parameters in age-related cataract patients", to make it more explicit for a wider audience.

We updated all the points concerning Journal Requirements. The authors received no specific funding for this work.

We have carefully considered and responded to all the points addressed by the reviewers. Per your instructions, all substantive amendments in the revised version are stated in our point-by-point response, and are marked in red in the article. 

We greatly hope that this new version will meet the reviewers' expectations and comply with your editorial policy. 

Yours sincerely,

Dr. Thibaud GARCIN, M.D., Ph.D., FEBO

t.garcin@univ-st-etienne.fr

Dear Dr. Garcin,

Thank you for submitting your manuscript to PLOS ONE. After careful consideration, we feel that it has merit but does not fully meet PLOS ONE’s publication criteria as it currently stands. Therefore, we invite you to submit a revised version of the manuscript that addresses the points raised during the review process.

We would appreciate receiving your revised manuscript by Jun 21 2020 11:59PM. To enhance the reproducibility of your results, we recommend that if applicable you deposit your laboratory protocols in protocols.io, where a protocol can be assigned its own identifier (DOI) such that it can be cited independently in the future. 

• A rebuttal letter that responds to each point raised by the academic editor and reviewer(s). This letter should be uploaded as separate file and labeled 'Response to Reviewers'.

• A marked-up copy of your manuscript that highlights changes made to the original version. This file should be uploaded as separate file and labeled 'Revised Manuscript with Track Changes'.

• An unmarked version of your revised paper without tracked changes. This file should be uploaded as separate file and labeled 'Manuscript'.

We look forward to receiving your revised manuscript.

Kind regards,

Ireneusz Grulkowski, PhD

Academic Editor

PLOS ONE

Journal Requirements:

Updates were done.

"NO

The funders had no role in study design, data collection and analysis, decision to

publish, or preparation of the manuscript."

No funding was received for this work.

“The authors received no specific funding for this work.”

Reviewers' comments:

Reviewer's Responses to Questions

Comments to the Author  1. Is the manuscript technically sound, and do the data support the conclusions?  The manuscript must describe a technically sound piece of scientific research with data that supports the conclusions. Experiments must have been conducted rigorously, with appropriate controls, replication, and sample sizes. The conclusions must be drawn appropriately based on the data presented. 

Reviewer #1: Partly

Reviewer #2: Yes

Reviewer #3: Partly

2. Has the statistical analysis been performed appropriately and rigorously? 

Reviewer #1: No

Reviewer #2: Yes

Reviewer #3: Yes

3. Have the authors made all data underlying the findings in their manuscript fully available?  The PLOS Data policy requires authors to make all data underlying the findings described in their manuscript fully available without restriction, with rare exception (please refer to the Data Availability Statement in the manuscript PDF file). The data should be provided as part of the manuscript or its supporting information, or deposited to a public repository. For example, in addition to summary statistics, the data points behind means, medians and variance measures should be available. If there are restrictions on publicly sharing data—e.g. participant privacy or use of data from a third party—those must be specified.

Reviewer #1: Yes

Reviewer #2: Yes

Reviewer #3: Yes

4. Is the manuscript presented in an intelligible fashion and written in standard English?  PLOS ONE does not copyedit accepted manuscripts, so the language in submitted articles must be clear, correct, and unambiguous. Any typographical or grammatical errors should be corrected at revision, so please note any specific errors here.

Reviewer #1: No

Reviewer #2: Yes

Reviewer #3: No

We have considered the corrections advised by Reviewers 1 & 3, and the manuscript has now been reviewed by a native English speaker.

 

5. Review Comments to the Author  Please use the space provided to explain your answers to the questions above. You may also include additional comments for the author, including concerns about dual publication, research ethics, or publication ethics. (Please upload your review as an attachment if it exceeds 20,000 characters)

The questions and comments of the 3 reviewers helped us to rewrite a significant part of the article in particular by simplifying the abstract, shortening the introduction and completing the results for the 3 OQAS parameters.

In addition, we would also like to slightly modify the title to make it more explicit for a wider audience.

The new title we propose is now : Using Optical Quality Analysis System (OQAS) for predicting surgical parameters in age-related cataract patients.

Reviewer #1: The study evaluated the association between OSI, LOCS III and phacodynamics, and there are some questions need to be considered.

1. The title used the Optical quality analysis system, but the author seems to focus on the OSI. The criterion was only related to OSI, so I am not clear why the relationships between MTF cutoff, LOCS III and phacodynamics were studied here. The abstract also did not mention the results of MTF cutoff. 

Initially we had chosen to present only OSI to simplify but your remark is correct. We have therefore decided to present all the results of the OQAS. The text and the abstract have been modified accordingly.

2. The English used in the paper need to be improved. Occasionally, it is difficult to understand some statements. A professional editing service is urgently needed. 

The manuscript has now been reviewed by a native English speaker.

3. The Introduction part is relatively long, and the author should simplify the text. 

We have shortened the introduction considerably.

4. The cataract grading score (LOCS III) used in the study were grade variables, which should not use mean±SD, and the correlation analysis should be changed to spearman correlation.

The LOCSIII score is perfectly comparable to a continuous variable as mentioned in princeps publication1, and used by numerous authors in different published studies in the field2-7. 

Each of the 4 items (Nuclear opacity/Nuclar Color/Cortical cataract/Posterior cataract) that compose it is defined to the decimal from 0.1 to 5.9 5C&P) or 6.9 (NC&NO). The overall score thus varies from 0.1 to 0.1 between 0.4 and 25.6. Consequently, mean±SD can be used for LOCSIII grading. We detailed the LOCSIII grading system in material and methods section for clarification. 

References:

1. Chylack LT, Jr., Wolfe JK, Singer DM, Leske MC, Bullimore MA, Bailey IL, et al. The Lens Opacities Classification System III. The Longitudinal Study of Cataract Study Group. Arch Ophthalmol. 1993;111(6):831-6. PubMed PMID: 8512486.

2. Karbassi M, Khu PM, Singer DM, Chylack LT, Jr. Evaluation of lens opacities classification system III applied at the slitlamp. Optometry and vision science: official publication of the American Academy of Optometry. 1993;70(11):923-8. Epub 1993/11/01. PubMed PMID: 8302528.

3. Wong WL, Li X, Li J, Cheng CY, Lamoureux EL, Wang JJ, et al. Cataract conversion assessment using lens opacity classification system III and Wisconsin cataract grading system. Invest Ophthalmol Vis Sci. 2013;54(1):280-7. Epub 2012/12/13. doi: 10.1167/iovs.12-10657. PubMed PMID: 23233255.

4. Cabot F, Saad A, McAlinden C, Haddad NM, Grise-Dulac A, Gatinel D. Objective assessment of crystalline lens opacity level by measuring ocular light scattering with a double-pass system. American journal of ophthalmology. 2013;155(4):629-35, 35 e1-2. doi: 10.1016/j.ajo.2012.11.005. PubMed PMID: 23317652.

5. Pan AP, Wang QM, Huang F, Huang JH, Bao FJ, Yu AY. Correlation among lens opacities classification system III grading, visual function index-14, pentacam nucleus staging, and objective scatter index for cataract assessment. American journal of ophthalmology. 2015;159(2):241-7 e2. Epub 2014/12/03. doi: 10.1016/j.ajo.2014.10.025. PubMed PMID: 25448993.

6. Wong AL, Leung CK, Weinreb RN, Cheng AK, Cheung CY, Lam PT, et al. Quantitative assessment of lens opacities with anterior segment optical coherence tomography. Br J Ophthalmol. 2009;93(1):61-5. Epub 2008/10/08. doi: 10.1136/bjo.2008.137653. PubMed PMID: 18838411.

7. Galliot F, Patel SR, Cochener B. Objective Scatter Index: Working Toward a New Quantification of Cataract? Journal of refractive surgery. 2016;32(2):96-102. doi: 10.3928/1081597X-20151222-02. PubMed PMID: 26856426.

For correlation, we therefore used Spearman or Pearson according to normality of the data distribution.

In addition, the author mentioned the regression analysis in the Figure legends, but not occurred in the main manuscript.

Thank you for that comment. We have added in M&M that linear regressions have been done.

And the independent variables and dependent variables should be defined in the whole study. 

The only independent variable is the grade of the cataract (LOCSIII score). A priori, all the others can be considered as dependent.

Logically, the hypotheses are that the parameters of the OQAS must depend on LOCSIII and that it is the same for the surgical parameters (CDE and US time being very good indicators of the hardness of the crystalline lens). However, the establishment of LOCSIII remains subjective and has never been automated since 1993. What we have therefore sought to determine is whether one or more objective parameters of the OQAS were correlated with the surgical parameters and could thus make it possible to predict the latter.

5. The author analyzed the correlation between OSI and LOCS III, LOCS III and phacodynamics, then analyze the association between OSI and phacodynamics. The Partial correlation need to be used and the LOCS III should be considered in the association between OSI and phacodynamics.

We thank you for this suggestion. Indeed, we have now calculated partial correlation coefficients between OQAS and surgical parameters, taking into account LOCSIII.

For paragraph results 

“Partial correlation between CDE and OSI, US time and OSI controlling for the LOCSIII was negligible (r = 0.013 and r = -0.041 respectively) and not significant (p=0.993 and p=0.910 respectively), thus indicating that LOCSIII N score greatly influenced the relation between OSI and surgical parameters.”

For paragraph discussion

“Partial correlation analysis clearly demonstrates that LOCSIII acts as a confounding factor to explain the strong correlation between OSI and phacodynamics. This is explained by the fact that LOCSIII and OQAS both measures the same physical phenomenon: lens opacification. This is a demonstration of OQAS reliability as objective and automatic measurement.”

6. The “95%CI” in Figure legends should be cleared.

We do not understand why you wanted us to remove the confidence intervals. We think it is necessary and honest considering our sample size. We precised in material and methods paragraph that “The Red dots show the 95% confidence interval (95%CI) of the slope and intercept on the linear regression graphs”. So everything is clearer and we don’t note iteratively the same thing for each linear regression graph.

Reviewer #2: Dear Thibaud Garcin,

First of all, I would like to thank for submitting in PLOS ONE your work “Using Optical quality analysis system for cataract preoperative assessment & phacodynamics prediction in age-related cataract patients”. I have to admit that this work might be very useful for ophthalmologists and consequently, it might improve the life quality of our society. However, there are some doubts, that I would like to ask you:

Material and Methods

a) Involved eligible patients.

• In relation to the topic “involved eligible patients”, it is indicated that all patients were under topical anaesthesia. Is it possible to indicate from which brand was used this anaesthesia?

We have now made that clear in the text.

One drop of Oxybuprocaine chlorhydrate 0,4% (Théa, Clermont-Ferrand, France). and one drop of Tetracaine 1% (Théa, Clermont-Ferrand, France) were instilled twice in the conjunctival sac, 5 minutes apart, 10 minutes before draping. 

• In this section, it might appropriate to include information about the subjects who participated in this clinical trial (initial number of patients and eyes, age and gender, refractive error, etc.) instead of the results section.

We moved these data from results section to material and methods section, and deleted figure 1 caption as well as figure 1 not so necessary considering already details in text.

b) Preoperative subjective and objective assessment.

• In relation to the gold-standard LOCSIII test, I would like to know the reason why the nuclear opacity (NO) and nuclear colour (NC) are scaled from 1 to 6 meanwhile the sub-capsular (SCP) and cortical cataract (C) are scaled from 1 to 5. Moreover, it might be very useful to indicate it in the manuscript.

The LOCSIII grading system was published by Chylack et al. in 19931 and is conceived like this, and this gold standard did not change. But often, authors referring to this grading tend to adapt their own “scale” by simplifying categories/range/scales2,3.

References:

1. Chylack LT, Jr., Wolfe JK, Singer DM, Leske MC, Bullimore MA, Bailey IL, et al. The Lens Opacities Classification System III. The Longitudinal Study of Cataract Study Group. Arch Ophthalmol. 1993;111(6):831-6. PubMed PMID: 8512486.

2. Karbassi M, Khu PM, Singer DM, Chylack LT, Jr. Evaluation of lens opacities classification system III applied at the slitlamp. Optometry and vision science: official publication of the American Academy of Optometry. 1993;70(11):923-8. Epub 1993/11/01. PubMed PMID: 8302528.

3. Wong WL, Li X, Li J, Cheng CY, Lamoureux EL, Wang JJ, et al. Cataract conversion assessment using lens opacity classification system III and Wisconsin cataract grading system. Invest Ophthalmol Vis Sci. 2013;54(1):280-7. Epub 2012/12/13. doi: 10.1167/iovs.12-10657. PubMed PMID: 23233255.

• OQAS measurements were done in both eyes where some subjects showed that one eye was undergone to phacoemulsification. This data allowed to check the reliable and consistent of the OQAS. It might be very interested to include these to show the advantage of the OQAS.

As the aim was to analyze the relationships between OQAS parameters, surgical parameters and LOCSIII we included in the statistical analysis only the operated eyes.

As you suggest, we have added, for your information, this descriptive paragraph below of the OQAS data of non-operated eyes, only this file and not in the manuscript to make it clearer and simpler.

“We used the data acquired for each fellow eye (phakic or pseudophakic) to check if the data concerning each operated eye was reliable and consistent.

For the fellow eyes, 2 were long-standing visually impaired (2.0 logMAR) with corneal scars, 14 were pseudophakic with median far BCVA 0.10 (0.00 – 0.30) logMAR and mean OSI 1.27 ±0.41 [0.5-2.1], 5 were phakic (3 pure early cortical cataracts rated LOCSIII, 2 pure early nuclear cataracts) with median far BCVA 0.2 (0.10 – 0.30) logMAR and mean OSI 5.54 ±1.83 [3.9-8.4]. The 5 patients with phakic fellow eye were not really bothered compared to their other eye operated and included for analysis.

Regarding the fellow eyes, OQAS provided reliable data for phakic and pseudophakic eyes with a strong correlation between BCVA and OSI: respectively Spearman r=0.797 p<0.001 and Spearman r=0.888 p=0.044.”

c) Statistical analysis

• Regarding to the statistical analysis section, it is not indicated which type of test is used to compare the proposal parameters. I would recommend to indicate that all parameters were compared by linear correlations in order to avoid misunderstanding with another tests like T – student test or ANOVA.

We reviewed the statistical analysis and clarified the methods used.

• In the line 210, it is indicated “mean + SD”. Is it possible to indicate in the manuscript what it means SD.? I think, it is Standard Deviation.

This has been done

Results

a) Baseline population data

• LOCSIII test was done by two observers and no differences were appreciated between them. I would like to ask you how it was analysed the reproducibility of the LOCSIII between these 2 observers to obtain a p-value = 0.51.

Thanks for your note. As LOCSIII is decimal score, the agreement between the 2 ophthalmologists for LOCSIII grading was determined by calculating the mean difference and the correlation coefficient. The mean LOCSIII total score was 4.86 ±2.03 with excellent agreement between observers (mean difference 0.3; r=0.94 p<0.001, 19 cases with perfect agreement, 3 cases with differences of 2 points).

We added the points in methods & results section.

• In table 1, is it possible to indicate which parameters are following a normal and not normal distribution after doing the Sapphire – Wilk test. In addition, in the title box Mean +- SD, it should be included median +- percentiles

We reconsidered the statistical analysis and, given the numbers involved, preferred to display more robust non-parametric tests. Nevertheless, we show in the table the data with a normal distribution (Shapiro-Wilk test and Kolmogorov-Smirnov test). We have performed in parallel the 2 types of tests, parametric and non-parametric. In 95% of cases, both tests gave the same result, which is a good indicator of robustness.

b) Correlations

• In relation to figure 2, I am quite surprised about the dispersion when it is correlated the BCVA with the OSI and with LOCSIII test. I am wondering if this dispersion might be associated to the low influence of the visual acuity in the decision to cataract surgery as it was indicated in the introduction. It would be fantastic to discuss this point in the discussion section.

In our sample of population, LOCSIII C is well represented with different grades, and it may impact on dispersion of data. OSI is more dispersed as well as BCVA when we consider Cortical components (as reported by Vilseca et al. 10.1136/bjophthalmol-2011-301055).

In our department the surgical indications for cataract surgery usually follow the national recommendations (= or >0.3 logMAR). 

We added this point to the discussion.

“We found moderate correlation between OSI and BCVA as other teams [51, 56], while other authors found a stronger correlation [21, 52, 59]. Differences can be explained by different visual acuity threshold for decision-making surgery between the different centers, and, as explained above for correlation between LOCSIII and BCVA, by the differences in proportion of nuclear and cortical cataracts [59], cortical components are well represented with different grades in our population.”

• Another interesting point that I would like to ask you in relation to the figure 3 is about the slopes of each graphic. It seems that the correlation LOCSIII C vs OSI shows the higher slope (with r=0.715, which it is the highest), but as well the most dispersive graphic (p-value = 0.02 although <0.05). How clinically might it be interpreted? It is a pity that the sample population is too low (n=10) to analyse better this behaviour. 

Despite the small size of our population with cortical cataract (n=10 indeed), we find results consistent with those of Vilaseca et al. : They found that BCVA and OSI are significantly more scattered in groups with cortical and posterior subcapsular cataract. They also showed that the best agreement between OSI and LOCSIII was found for cortical cataracts. (Vilaseca M, Romero MJ, Arjona M, Luque SO, Ondategui JC, Salvador A, et al. Grading nuclear, cortical and posterior subcapsular cataracts using an objective scatter index measured with a double-pass system. (Br J Ophthalmol. 2012;96(9):1204-10. doi: 10.1136/bjophthalmol-2011-301055. PubMed PMID: 22790434.).

Moreover, according to these graphics, it is observed that the scattering is higher in nuclear cataracts, but the strong OSI change is caused in the cortical cataracts? Do you know why?

It is demonstrated that cortical cataract influences OSI more than nuclear cataract as explained above. Dispersion is not greater in the nuclear group. We have redrawn all the diagrams with the same scales to facilitate visual comparisons.

Here one part of the new graph, with same scale on X axis.

Regarding to figure 4, how is it possible that the CDE hasn´t got any influence in the cortical cataract? According to your correlations, it is not significance (p-value = 0.058). I think, that it is caused by the sample population n=10. What do you think?

The cortex always remains soft even when it becomes opaque. Regardless of the level of opacity the cortex can always be extracted without the use of ultrasound, with an irrigation aspiration cannula. It is therefore completely logical not to find any correlation between cortical LOCSIII and CDE or US Time. Therefore, a larger number of patients would probably not change anything. There are 2 other articles that did not find a relationship between LOCSIIIC and CDE or US Time 1,2.

References

1. Davison JA, Chylack LT. Clinical application of the lens opacities classification system III in the performance of phacoemulsification. Journal of cataract and refractive surgery. 2003;29(1):138-45. Epub 2003/01/29. PubMed PMID: 12551681.

2. Bencic G, Zoric-Geber M, Saric D, Corak M, Mandic Z. Clinical importance of the lens opacities classification system III (LOCS III) in phacoemulsification. Collegium antropologicum. 2005;29 Suppl 1:91-4. Epub 2005/10/01. PubMed PMID: 16193685.

• According to figure 6, I think it does not have any sense to include it because the MTF and OSI are specific parameters from OQAS which are strongly related between them by the PSF. Thus, the OQAS must give strong correlation to have a high reproducibility. Consequently, this correlation does not give any significant information for a clinical purpose. I would recommend to remove the data and graphics between MTF and OSI and prepare another figure with correlations between MTF with BCVA, MTF with LOCSIII, MTF with CDE and MTF with US.

We followed your suggestion and deleted fig 6. Instead we added a graph for relationship between MTF and clinical and surgical parameters (and the same for SR as suggested in specifics comments)

Discussion

• In the discussion about the BCVA vs OQAS, it is explained that the differences between your research with other ones might be caused by the sample size or the visual acuity decision-making surgery after consulting people. However, there is not any references about the type of visual test was used and how it might impact in the correlated results.

French ophthalmologists use the decimal Monoyer chart for routine far vision. This was added in the revised text. Visual acuity was converted in LogMar to allow performing statistics.

Reviewer #3: COMMENT:

This article discusses the usefulness of the OSI parameter in the clinical management of age-related cataracts. Furthermore, the relationship of preoperative parameters with photodynamic parameters involved in surgery is interesting and original. However, the sample of patients included in the study is very small. 

In addition, as several works recommend, the statistical analysis must be corrected to include 1 eye per patient. In fact, only one case in this work does not meet this criterion and the results of the study are unlikely to be different if the correction is made. 

We have updated all statistical analysis on 21 eyes, with one eye per subject. As expected the results and their interpretation did not change.

Likewise, the discussion should be improved for a better understanding.

The discussion was rewritten and simplified

For example, when the author removes cases with a cortical component, he consequently removes cases with OSI> 9. This should be added in the discussion. 

We added comment in discussion.

“Therefore, After removing cortical and sub capsular components, for pure nuclear cataracts only, OSI ranged from 1.4 to 8.3 and strong correlations were found between CDE and OSI. The correlation was even stronger than between CDE and LOCSIII N, suggesting that OQAS could better predict phacodynamics than LOCSIII for pure nuclear cataracts. The same findings were found for US time and OSI, or US time and LOCSIII N.”

The influence of aberrations on the overestimation of OSI should be discussed as well.

This time we have integrated the detailed analysis of the SR parameter which is the most sensitive to high-order aberrations.

“We found also strong significant correlations between: 1/ Strehl ratio and OSI (rs = -0.948, p<0.001); 2/ BCVA and Strehl ratio (rs = -0.622, p=0.003); 3/ LOCSIII total and Strehl ratio (rs = -0.676, p<0.001) as well as in subgroups of different LOCSIII components (N, NC, NO) except for LOCSIII C (p=0.056); 4/ Strehl ratio and CDE (Spearman r = -0.664, p=0.031), and also Strehl ratio and US time (Spearman r = -0.653, p=0.032) only for pure nuclear cataracts (Fig.6).”

As far as language and style are concerned, the use of punctuation marks does need some revising as on certain occasions it interferes with clarity. See, for example, lines 341 – 345 and 402 - 403. Similarly, the paper shows some segments lacking in clear sentence structure that should be rewritten (lines 21 – 23; 159 -163; 207 – 208; 222 – 224). Other aspects that should be revised include unnecessary use of passive forms - particularly of the verb “correlate” -, the use of contractions, vocabulary (“similar as” in lines 364 – 385 should be “similar to”), missing words (“decision-making surgery”) and verb tense consistency (lines 160 – 161). 

The manuscript has now been reviewed by a native English speaker.

SPECIFIC COMMENTS:

Page 8. Lines 159-163: this sentence should be rewritten for a better understanding. 

Sentence was rewritten as suggested by reviewer N°3.

“We used the data acquired for each fellow eye (phakic or pseudophakic) to check if the data concerning each operated eye was reliable and consistent.“

Page 9. Lines 180-183: the definition of SR does not exactly correspond to the one used by the double-pass system, which computes the SR in two dimensions as the ratio between the areas under the MTF curve of the measured eye and that of the aberration-free eye. Ref.: DOI:10.1111/j.1444-0938.2010.00535.x 

Thank you for this comment. We updated the definition.

Page 9. Lines 194-195: missing parenthesis open and close.

Missing parenthesis were added.

Page 11. Line 225: in the statistical analysis it is recommended to include an eye per patient in all cases due to strong correlation between the two eyes of a subject.

We updated all statistical analysis on 21 eyes, with one eye per subject.

Page 14. Lines 298-303: correlations with SR should be included in the results since this parameter is the one most related to loss of image quality and very sensitive to aberrations. 

We added data concerning SR in results and discussion sections.

“We found also strong significant correlations between: 1/ Strehl ratio and OSI (rs = -0.948, p<0.001); 2/ BCVA and Strehl ratio (rs = -0.622, p=0.003); 3/ LOCSIII total and Strehl ratio (rs = -0.676, p<0.001) as well as in subgroups of different LOCSIII components (N, NC, NO) except for LOCSIII C (p=0.056); 4/ Strehl ratio and CDE (Spearman r = -0.664, p=0.031), and also Strehl ratio and US time (Spearman r = -0.653, p=0.032) only for pure nuclear cataracts (Fig.6).”

Page 14. Line 305: correlation between the MTF and OSI is evident, especially considering that they have been obtained with the same instrument. Therefore, Figure 6 is unnecessary.

We deleted this Figure 6 (of the R0 version) for the R1 version.

---

## [Decision Letter · Decision Letter 1]

30 Jul 2020

PONE-D-20-08844R1

Using Optical Quality Analysis System for predicting surgical parameters in age-related cataract patients

PLOS ONE

Dear Dr. Garcin,

Thank you for submitting your manuscript to PLOS ONE. After careful consideration, we feel that it has merit but does not fully meet PLOS ONE’s publication criteria as it currently stands. Therefore, we invite you to submit a revised version of the manuscript that addresses the points raised during the review process.

We look forward to receiving your revised manuscript.

Kind regards,

Ireneusz Grulkowski, PhD

Academic Editor

PLOS ONE

Journal Requirements:

Additional Editor Comments (if provided):

Please, address additional comments of the reviewers.

Reviewers' comments:

Reviewer's Responses to Questions

**Comments to the Author**

1. If the authors have adequately addressed your comments raised in a previous round of review and you feel that this manuscript is now acceptable for publication, you may indicate that here to bypass the “Comments to the Author” section, enter your conflict of interest statement in the “Confidential to Editor” section, and submit your "Accept" recommendation.

Reviewer #1: All comments have been addressed

Reviewer #2: All comments have been addressed

Reviewer #3: All comments have been addressed

2. Is the manuscript technically sound, and do the data support the conclusions?

Reviewer #1: Yes

Reviewer #2: Partly

Reviewer #3: Yes

3. Has the statistical analysis been performed appropriately and rigorously? 

Reviewer #1: Yes

Reviewer #2: Yes

Reviewer #3: Yes

4. Have the authors made all data underlying the findings in their manuscript fully available?

Reviewer #1: Yes

Reviewer #2: Yes

Reviewer #3: Yes

5. Is the manuscript presented in an intelligible fashion and written in standard English?

Reviewer #1: Yes

Reviewer #2: No

Reviewer #3: Yes

6. Review Comments to the Author

Reviewer #1: (No Response)

Reviewer #2: Dear Thibaud Garcin,

Thank you for submitting in PLOS ONE your research work described in the manuscript “Using Optical Quality Analysis System for predicting surgical parameters in age-related cataract patients”. The new version of the manuscript has improved considerable in clearly and comprehension terms. However, I would like to comment some small details that I have been able to observe in this review:

• First of all, I would like to suggest a deep revision of the manuscript in order to improve some expressions and correct grammar mistakes. Although, in general terms the text is comprehensible, there are some sections where there are incoherent expressions and not well-written. I do recommend to send the manuscript to an expert or native speaker to improve the writing quality.

Introduction

• I think that the definition of the Strehl ratio (SR) is not correct. The SR does not depend on the Modulate Transfer Function (MTF). The SR is the ration between the peak intensity from the Point Spread Function (PSF) of the aberrated eye and the peak intensity from the PSF of the non-aberrated eye. Please, correct lines 113 to 115.

Material and Methods

a) Patients and ethic statement

• The 2 last paragraph are not well located in this section and it makes confusion to the reader. In lines 132 – 135, it is described that the only inclusion criteria is to obtain good OSI measurements. However, it is not known any exclusion criteria and ophthalmic exam until next section. Similarly, it is observed in the last paragraph (lines 136 – 141), where some subjects were excluded because of poor cooperation and dense cataracts, without knowing any details from phacodynamics mode adjustment. I would recommend to move this section before the section “statistical analysis” to describe previously inclusion and exclusion methods.

Results

a) Correlation Analysis: Main objective, subjective preoperative parameters and phacodynamics.

• In order to avoid misunderstanding, I do recommend strongly to indicate “LogMar scale” in the axis Best-Corrected Visual Acuity (BCVA) in figures 1, 5 and 6. The current values can confuse to the reader with decimal scale and the interpretation of the Spearman coefficient can be opposite.

b) Correlation Analysis: Other OQAS parameters, subjective preoperative parameters and phacodynamics.

• Relations between Optical Scattering Index (OSI) with Modulate Transfer Function (MTF) in lines 273 – 274; and between Strehl ratio (SR) with MTF in lines 281 – 282, should not be included in the manuscript. These correlations were well developed and thoroughly analysed by the company in order to determine the performance of the device. Consequently, these data are not going to give any particular and novel information.

Discussion

• I do not really agree with the idea that OQAS and LOCSIII measures the same physical phenomenon. Specially, because OQAS measures the quality of the whole eye, and a simple uveitis or a corneal scar can affect completely the measurement. I would be more specific in this sentence (lines 390 – 392) explaining that both methods will be measuring the same parameters if there are no more ocular pathologies.

Reviewer #3: Thank you for your answers and comments. You have solved most of my concerns successfully. Regarding language, it has improved significantly and it makes the text much easier to follow now. On the other hand, I personally think that IQR=Q-Q3 provides an easier explanation for the data dispersion and allows the comparison with SD (line 187 and tables).

Some specific comments:

- Line 90: Repetition of a fragment

- Lines 210 – 211: The sentence needs revising for a better understanding.

- Lines 240 – 242: Lack of statistical significance for correlation between LOCSIII C and CDE as well as US is probably due to insufficient number of patients in this subgroup (LOCSIII C). Notice that rs is > 0,580. You should comment it here and explain it in discussion.

- Figures 5 and 6: The top right and bottom left graphs seem to be wrong e.g. n = 21?

- I would be very grateful if you could send me the data rs and p for the different subgroups of LOCSIII components because note (lines 235 – 237) that they are not shown in Figure 1.

7. PLOS authors have the option to publish the peer review history of their article (what does this mean?). If published, this will include your full peer review and any attached files.

Reviewer #1: No

Reviewer #2: No

Reviewer #3: No

---

## [Author Response · Author response to Decision Letter 1]

29 Aug 2020

Saint-Etienne, August 29th 2020

Dear Dr. Grulkowski,

Please find attached a revised version R2 of our article "Using Optical Quality Analysis System for predicting surgical parameters in age-related cataract patients".

We updated all the points concerning Journal Requirements. The authors received no specific funding for this work.

We have carefully considered and responded to all the points addressed by the reviewers. 

We greatly hope that this new version will meet the reviewers' expectations and comply with your editorial policy. 

Yours sincerely,

Dr. Thibaud GARCIN, M.D., Ph.D., FEBO

t.garcin@univ-st-etienne.fr

PONE-D-20-08844R1

Using Optical Quality Analysis System for predicting surgical parameters in age-related cataract patients

PLOS ONE

Dear Dr. Garcin,

Thank you for submitting your manuscript to PLOS ONE. After careful consideration, we feel that it has merit but does not fully meet PLOS ONE’s publication criteria as it currently stands. Therefore, we invite you to submit a revised version of the manuscript that addresses the points raised during the review process.

We look forward to receiving your revised manuscript.

Kind regards,

Ireneusz Grulkowski, PhD

Academic Editor

PLOS ONE

Journal Requirements:

Additional Editor Comments (if provided):

Please, address additional comments of the reviewers.

Reviewers' comments:

Reviewer's Responses to Questions

Comments to the Author

1. If the authors have adequately addressed your comments raised in a previous round of review and you feel that this manuscript is now acceptable for publication, you may indicate that here to bypass the “Comments to the Author” section, enter your conflict of interest statement in the “Confidential to Editor” section, and submit your "Accept" recommendation.

Reviewer #1: All comments have been addressed

Reviewer #2: All comments have been addressed

Reviewer #3: All comments have been addressed

2. Is the manuscript technically sound, and do the data support the conclusions?

Reviewer #1: Yes

Reviewer #2: Partly

Reviewer #3: Yes

3. Has the statistical analysis been performed appropriately and rigorously?

Reviewer #1: Yes

Reviewer #2: Yes

Reviewer #3: Yes

4. Have the authors made all data underlying the findings in their manuscript fully available?

Reviewer #1: Yes

Reviewer #2: Yes

Reviewer #3: Yes

5. Is the manuscript presented in an intelligible fashion and written in standard English?

Reviewer #1: Yes

Reviewer #2: No

Reviewer #3: Yes

6. Review Comments to the Author

Reviewer #1: (No Response) 

Reviewer #2: Dear Thibaud Garcin,

Thank you for submitting in PLOS ONE your research work described in the manuscript “Using Optical Quality Analysis System for predicting surgical parameters in age-related cataract patients”. The new version of the manuscript has improved considerable in clearly and comprehension terms. However, I would like to comment some small details that I have been able to observe in this review:

• First of all, I would like to suggest a deep revision of the manuscript in order to improve some expressions and correct grammar mistakes. Although, in general terms the text is comprehensible, there are some sections where there are incoherent expressions and not well-written. I do recommend to send the manuscript to an expert or native speaker to improve the writing quality. The manuscript has now been reviewed again, this time by a native English speaker.

Introduction

• I think that the definition of the Strehl ratio (SR) is not correct. The SR does not depend on the Modulate Transfer Function (MTF). The SR is the ration between the peak intensity from the Point Spread Function (PSF) of the aberrated eye and the peak intensity from the PSF of the non-aberrated eye. Please, correct lines 113 to 115. Thank you for this comment. We updated the definition.

Material and Methods

a) Patients and ethic statement

• The 2 last paragraph are not well located in this section and it makes confusion to the reader. In lines 132 – 135, it is described that the only inclusion criteria is to obtain good OSI measurements. However, it is not known any exclusion criteria and ophthalmic exam until next section. Similarly, it is observed in the last paragraph (lines 136 – 141), where some subjects were excluded because of poor cooperation and dense cataracts, without knowing any details from phacodynamics mode adjustment. I would recommend to move this section before the section “statistical analysis” to describe previously inclusion and exclusion methods. Thank you for this comment. We create an “inclusion & exclusion criteria” section just before statistical analysis section.

Results

a) Correlation Analysis: Main objective, subjective preoperative parameters and phacodynamics.

• In order to avoid misunderstanding, I do recommend strongly to indicate “LogMar scale” in the axis Best-Corrected Visual Acuity (BCVA) in figures 1, 5 and 6. The current values can confuse to the reader with decimal scale and the interpretation of the Spearman coefficient can be opposite. Thank you for this comment. We have updated figures 1, 5 and 6.

b) Correlation Analysis: Other OQAS parameters, subjective preoperative parameters and phacodynamics.

• Relations between Optical Scattering Index (OSI) with Modulate Transfer Function (MTF) in lines 273 – 274; and between Strehl ratio (SR) with MTF in lines 281 – 282, should not be included in the manuscript. These correlations were well developed and thoroughly analysed by the company in order to determine the performance of the device. Consequently, these data are not going to give any particular and novel information. Thank you for this comment. The data in question have been deleted.

Discussion

• I do not really agree with the idea that OQAS and LOCSIII measures the same physical phenomenon. Specially, because OQAS measures the quality of the whole eye, and a simple uveitis or a corneal scar can affect completely the measurement. I would be more specific in this sentence (lines 390 – 392) explaining that both methods will be measuring the same parameters if there are no more ocular pathologies. Thank you for this comment. The clarification has been made.

Reviewer #3: Thank you for your answers and comments. You have solved most of my concerns successfully. Regarding language, it has improved significantly and it makes the text much easier to follow now. On the other hand, I personally think that IQR=Q-Q3 provides an easier explanation for the data dispersion and allows the comparison with SD (line 187 and tables). Thank you for this comment.

Some specific comments:

- Line 90: Repetition of a fragment. Thank you for this comment. The repetition has been removed.

- Lines 210 – 211: The sentence needs revising for a better understanding. The sentence was revised.

- Lines 240 – 242: Lack of statistical significance for correlation between LOCSIII C and CDE as well as US is probably due to insufficient number of patients in this subgroup (LOCSIII C). Notice that rs is > 0,580. You should comment it here and explain it in discussion. As section results must be factual, we have now explained it in the discussion, per your suggestion.

- Figures 5 and 6: The top right and bottom left graphs seem to be wrong e.g. n = 21? Thank you for this comment. The titles of the top-right and bottom-left graphs were inverted. This mistake has now been corrected.

- I would be very grateful if you could send me the data rs and p for the different subgroups of LOCSIII components because note (lines 235 – 237) that they are not shown in Figure 1.

We found significant correlations between BCVA and LOCSIII total (rs = 0.561 ; p=0.008 ; Y = 6.258*X + 2.575) (Fig 1) as well as in subgroups of different LOCSIII components 

N rs = 0.633 ; p=0.005 ; Y = 4.738*X + 2.678

NC rs = 0.621 ; p=0.006 ; Y = 2.294*X + 1.338

NO rs = 0.581 ; p=0.012 ; Y = 2.444*X + 1.340

except for LOCSIII C rs = 0.007 ; p=0.696 ; Y = -0.746*X + 2.284

7. PLOS authors have the option to publish the peer review history of their article (what does this mean?). If published, this will include your full peer review and any attached files.

Do you want your identity to be public for this peer review? For information about this choice, including consent withdrawal, please see our Privacy Policy.

Reviewer #1: No

Reviewer #2: No

Reviewer #3: No

---

## [Decision Letter · Decision Letter 2]

25 Sep 2020

Using Optical Quality Analysis System for predicting surgical parameters in age-related cataract patients

PONE-D-20-08844R2

Dear Dr. Garcin,

We’re pleased to inform you that your manuscript has been judged scientifically suitable for publication and will be formally accepted for publication once it meets all outstanding technical requirements.

Kind regards,

Ireneusz Grulkowski, PhD

Academic Editor

PLOS ONE

Additional Editor Comments (optional):

Please address the comment of the 3rd reviewer during proof.

Reviewers' comments:

Reviewer's Responses to Questions

**Comments to the Author**

1. If the authors have adequately addressed your comments raised in a previous round of review and you feel that this manuscript is now acceptable for publication, you may indicate that here to bypass the “Comments to the Author” section, enter your conflict of interest statement in the “Confidential to Editor” section, and submit your "Accept" recommendation.

Reviewer #1: All comments have been addressed

Reviewer #2: All comments have been addressed

Reviewer #3: All comments have been addressed

2. Is the manuscript technically sound, and do the data support the conclusions?

Reviewer #1: Yes

Reviewer #2: Yes

Reviewer #3: Yes

3. Has the statistical analysis been performed appropriately and rigorously? 

Reviewer #1: Yes

Reviewer #2: Yes

Reviewer #3: Yes

4. Have the authors made all data underlying the findings in their manuscript fully available?

Reviewer #1: Yes

Reviewer #2: Yes

Reviewer #3: Yes

5. Is the manuscript presented in an intelligible fashion and written in standard English?

Reviewer #1: Yes

Reviewer #2: Yes

Reviewer #3: Yes

6. Review Comments to the Author

Reviewer #1: the study uses Optical Quality Analysis System to predicte surgical parameters in age-related cataract patients.

Reviewer #2: (No Response)

Reviewer #3: Thank you for your satisfying answers and comments. The new version of the manuscript has improved considerably. However, I would like to comment on a small detail that I have been able to observe in this review:

In the Introduction (lines 97-99) the last sentence in the paragraph states, ““ Scheimpflug camera or Shack-Hartman technology do not consider light scattering [33] and can overestimate the optical quality of an image when scattering affect the eyes, for example in cataract [34].”

This statement is true for Shack-Hartmann technology (by the way, the spelling of the surname needs correcting) but it is not for the Scheimpflug camera. I suggest removing the Scheimpflug camera from the sentence.

7. PLOS authors have the option to publish the peer review history of their article (what does this mean?). If published, this will include your full peer review and any attached files.

Reviewer #1: No

Reviewer #2: No

Reviewer #3: No

---

## [Editor Report · Acceptance letter]

1 Oct 2020

PONE-D-20-08844R2 

Using Optical Quality Analysis System for predicting surgical parameters in age-related cataract patients 

Dear Dr. Garcin:

I'm pleased to inform you that your manuscript has been deemed suitable for publication in PLOS ONE. Congratulations! Your manuscript is now with our production department. 

Kind regards, 

on behalf of

Dr. Ireneusz Grulkowski 

Academic Editor

PLOS ONE